# Transcriptional Profiling of the Small Intestine and the Colon Reveals Modulation of Gut Infection with *Citrobacter rodentium* According to the Vitamin A Status

**DOI:** 10.3390/nu14081563

**Published:** 2022-04-08

**Authors:** Zhi Chai, Yafei Lyu, Qiuyan Chen, Cheng-Hsin Wei, Lindsay M. Snyder, Veronika Weaver, Aswathy Sebastian, István Albert, Qunhua Li, Margherita T. Cantorna, Catharine Ross

**Affiliations:** 1Intercollege Graduate Degree Program in Physiology, The Pennsylvania State University, University Park, PA 16802, USA; zhichaizju@gmail.com; 2Department of Nutritional Sciences, The Pennsylvania State University, University Park, PA 16802, USA; qchen@biomagneticsolutions.com (Q.C.); ginawei420@gmail.com (C.-H.W.); 3Intercollege Graduate Degree Program in Bioinformatics and Genomics, The Pennsylvania State University, University Park, PA 16802, USA; lyuyafei@gmail.com; 4Department of Veterinary and Biomedical Sciences, The Pennsylvania State University, University Park, PA 16802, USA; lmsnyder@unm.edu (L.M.S.); vcw100@psu.edu (V.W.); mxc69@psu.edu (M.T.C.); 5Huck Institutes of the Life Sciences, The Pennsylvania State University, University Park, PA 16802, USA; azs13@psu.edu; 6Department of Biochemistry and Molecular Biology, The Pennsylvania State University, University Park, PA 16802, USA; iua1@psu.edu; 7Department of Statistics, The Pennsylvania State University, University Park, PA 16802, USA; qul12@psu.edu

**Keywords:** vitamin A, diarrhea, *Citrobacter rodentium*, RNAseq, gene expression, regulatory pathway, ion absorption, water loss

## Abstract

Vitamin A (VA) deficiency and diarrheal diseases are both serious public health issues worldwide. VA deficiency is associated with impaired intestinal barrier function and increased risk of mucosal infection-related mortality. The bioactive form of VA, retinoic acid, is a well-known regulator of mucosal integrity. Using *Citrobacter rodentium*-infected mice as a model for diarrheal diseases in humans, previous studies showed that VA-deficient (VAD) mice failed to clear *C. rodentium* as compared to their VA-sufficient (VAS) counterparts. However, the distinct intestinal gene responses that are dependent on the host’s VA status still need to be discovered. The mRNAs extracted from the small intestine (SI) and the colon were sequenced and analyzed on three levels: differential gene expression, enrichment, and co-expression. *C. rodentium* infection interacted differentially with VA status to alter colon gene expression. Novel functional categories downregulated by this pathogen were identified, highlighted by genes related to the metabolism of VA, vitamin D, and ion transport, including improper upregulation of Cl^−^ secretion and disrupted HCO_3_^−^ metabolism. Our results suggest that derangement of micronutrient metabolism and ion transport, together with the compromised immune responses in VAD hosts, may be responsible for the higher mortality to *C. rodentium* under conditions of inadequate VA.

## 1. Introduction

Diarrheal disease causes ~0.8 million deaths per year in children under the age of five and ranks as the second leading cause of infection-related mortality in this demographic group [1,2]. Recurrent early childhood diarrhea, occurring during the period of rapid growth of the brain and other organ systems, can contribute to lasting impairment in fitness, growth, and cognition. According to a report of Petri et al. [3], repeated diarrhea in the first two years of life was associated with a loss of 10 IQ points and one year of schooling by nine years of age. Diarrhea itself is caused by an imbalance of absorption and secretion of ions and solutes in the intestine, leading to rapid fluid loss and, potentially, lethal dehydration [4]. The introduction and widespread use of a nutritional treatment–oral rehydration therapy significantly reduced the global mortality from diarrheal diseases, from nearly 4.6 million per year in 1982 to 2.5 million per year by 2000 [5]. 

Numerous pathogens can cause diarrheal disease. Among the most studied is the enteropathogen *Escherichia coli* (*E. coli*), including enterohemorrhagic *E. coli* (EHEC) and enteropathogenic *E. coli* (EPEC), two related human pathogens which produce attaching/effacing (A/E) lesions of the intestinal epithelium and are major contributors to diarrhea-related deaths in children [6]. To study mechanisms of EHEC and EPEC, a mouse model of gut infection caused by *Citrobacter rodentium* (*C. rodentium*) has often been used. *C. rodentium* is a natural mouse pathogen, a Gram-negative bacterium that causes transmissible colonic hyperplasia and shares a core set of virulence factors with EPEC. Therefore, *C. rodentium* infection in mice is widely used to model the effects of EHEC and EPEC infection in humans [7]. In infected mice, *C. rodentium* first undergoes virulence changes, allowing the bacterium to colonize the colon and the rectum [8,9]. The peak of fecal shedding of *C. rodentium* takes place between days 7 and 14 post-infection (p.i.) and is accompanied by immune cell filtration, crypt hyperplasia, and diarrhea [10].

Clearance of *C. rodentium* infection requires both innate immunity (e.g., epithelial cells (EC), mucin production, apoptosis, antigen-presenting cells, and innate lymphoid cells) and adaptive immunity (i.e., Th1, Th17, Treg, and B cells). ECs, which constitute the first line of defense in the gut, become activated by the attachment of *C. rodentium* and thence produce factors including reactive oxygen species (ROS), antimicrobial peptides (AMP), and serum amyloid A (SAA) [11,12]. Epithelial hyperplasia and apoptosis are induced and may accelerate EC sloughing [13,14]. Epithelial goblet cells and their mucin products also play important roles in the pathogen clearance [15,16,17,18,19,20,21,22]. Additionally, *C. rodentium* infection induces a robust colonic Th17 response, which is essential for the clearance of this A/E pathogen [23,24,25]. Interleukins IL-17 and IL-22 produced by intraepithelial innate lymphoid cells 3 (ILC3) are essential factors to protect the gut against the infection at early stages, whereas a robust Th17 response is required at the later stages to clear the infection [26,27]. The susceptibility to diarrhea has been attributed to the disruption of water channels [28,29] and ion transporters relevant to sodium [28,29], potassium [28,29,30], and bicarbonates [18,28,29].

Micronutrients, especially vitamin A (VA) and vitamin D (VD), have long been known as modulators of mucosal immunity. Calcitriol, the active form of VD, is known as a potent immunosuppressor in the intestines [31,32,33,34,35]. Conversely, the metabolism of VA and VD is affected by infection [36,37,38,39,40,41]. 

According to a World Health Organization (WHO) report, diarrhea-related mortality has been reduced by 33% by VA supplementation [42]. Retinoic acid (RA) is the bioactive form of VA. In mice, RA functions in the education of mucosal dendritic cells (DCs), the differentiation of lymphocytes, and the imprinting of immune cells with gut-homing properties while also regulating the balance between the tolerance and the immunity, ILCs population, and the microbiota composition [43]. Mice that differ in the VA status also respond differently to the *C. rodentium* infection. Mice with the VA sufficient (VAS) status can thoroughly clear the *C. rodentium* infection within 4 weeks and recover from the infection completely. In contrast, VA-deficient (VAD) hosts suffered from a 40% mortality rate. The survivors failed to completely clear the pathogen and became non-symptomatic carriers that continuously transmit the disease [23]. Oral supplementation of RA can induce the pathogen clearance in VAD animals [23,25]. To sum up, reflected by the host survival rate, the pathogen clearance, and the colonic IL-17 production, an interaction between the VA status and the *C. rodentium* infection has been observed repeatedly by us and other investigators. 

Previously, we used an RNAseq approach to characterize the effects of the VA status alone in the small intestine (SI) and the colon [44]. Our focus on VA has public health relevance because VA deficiency remains a public health issue in resource-limited areas coincident with diarrhea prevalence. VA deficiency impairs the development, vision, immune functions, and is associated with the infectious disease severity [42]. It is estimated that VA deficiency impacts over 250 million preschool-aged children [45]. In the present study, we extended the current knowledge by seeking to identify the potential key mediators responsible for the resistance of VAS mice to the *C. rodentium* infection and the effects of interaction between the infection and the VA status on the global gene expression profiles in both the colon and the SI. Because the colon is a major site of the *C. rodentium* infection and the SI is a major site of VA metabolism, antigen presentation, and immune cell maturation, RNAseq was performed using both organs. We used the RNAseq analysis and a bioinformatics approach as a global and unbiased means to identify the genes and pathways affected by *C. rodentium* and as modified by the VA nutritional status. We hypothesized that the characterization of differential expression, co-expression networks, and enrichment analyses would identify the key genes and pathways involved in diarrheal disease that interact with the VA status. 

## 2. Materials and Methods

### 2.1. Animals

C57BL/6 mice were bred and maintained according to the guideline of the Institutional Animal Care and Use Committee (IACUC) at Pennsylvania State University. The mice were exposed to a 12 h dark/light cycle with continuous access to water and food. VA-sufficient (VAS) and VA-deficient (VAD) mice were generated as described previously [23,25,46,47] by feeding VAD or VAS diets to pregnant dams and weanlings. The VAD diet contained no VA, whereas the VAS diet provided 25 μg of retinyl acetate per day. The diet composition table is provided in the Appendix A [48]. After weaning, the mice were assigned to one of the four treatment groups (non-infected VAD, infected VAD, non-infected VAS, and infected VAS) until the end of the experiments. The VA status of the animals was confirmed by analyzing serum retinol using an ultra-performance liquid chromatography method [44].

### 2.2. Infection of C. rodentium

The origin, culture condition, and inoculation of *C. rodentium* have been described before [24]. Briefly, *C. rodentium* were grown overnight in a Difco Luria–Bertani (LB) broth containing 50 μg/mL nalidixic acid. The mice (8–10 weeks of age, individually housed) were fasted overnight prior to oral gavage with 5 × 10^9^ CFU of *C. rodentium*. To quantify the bacterial burdens, fresh fecal samples were collected thrice a week, homogenized, and plated on LB agar plates containing nalidixic acid. 

### 2.3. Tissue Collection and RNA Extraction

Two studies were conducted; one focused on the SI and one—on the colon, where the study was ended and tissues were collected on post-infection (p.i.) day 5 and day 10, respectively, the time of peak infection as shown in previous studies [23,25,38]. Details on tissue collection and RNA extraction were described previously [44]. Briefly, total RNA was extracted with a Qiagen RNeasy Midi Kit, genomics DNA was removed by a TURBO DNA-free kit, and RNA quality was determined on an Agilent Bioanalyzer to ensure RNAseq libraries were constructed with RNA samples of sufficient quality (RNA integrity number, RIN > 8).

### 2.4. RNAseq Library Preparation, Sequencing, and Mapping

Details on the methods of library preparation, sequencing, and mapping were provided before [44]. In brief, a TruSeq Stranded mRNA Library Prep kit and a HiSeq 2500 system (Illumina) were used to generate raw sequencing data. Quality trimming, mapping, coverage, and raw read counts were obtained with the following tools: trimmomatic [49], hisat2 alignment program [50], bedtools [51], and featureCounts [52]. 

### 2.5. Differential Expression

We removed low-expression transcripts from the raw count data matrix (Figure 1a). Then, differential expression (DE) analyses were performed using the DESeq2 package [53]. For the SI study, the transcript was removed if it had less than 10 counts in more than 8 of the 12 samples or if its total count was less than 200. For the colon study, the transcript was removed if it had less than 10 counts in more than 10 of the 16 samples or if its total count was less than 220 [44]. Differential expression analyses were performed and the 2 × 2 experimental design was resolved into three effects related to infection (Figure 1b): (I) the “VAS-Inf” effect, which corresponded to the comparison between the non-infected VAS verses the infected VAS groups; (II) the “VAD-Inf” effect, which corresponded to the comparison between the non-infected VAD and the infected VAD groups; and (III) “VA effect under infection”, which compared the infected VAD and the infected VAS groups. Criteria for differentially expressed genes (DEGs) were set at adjusted *p* (padj) < 0.05 and |FoldChange| > 2. The stringent criteria of DEG may lead to a large number of unchanged genes, i.e., genes that are neither upregulated (Log2FoldChange > 1, padj < 0.05) nor downregulated DEG (Log2FoldChange < −1, padj < 0.05). Visualizations such as heatmaps and volcano plots were created via R packages pheatmap, ggplot, and ggrepel. 

As a criterion for the DEGs corresponding to the interaction effect, padj < 0.05 was used, corresponding to the interaction effect. By definition, the interaction effect is the “difference on top of difference”. In the current study, the interaction effect examined how the infection effects differed between the different VA statuses. In other words, the interaction effect reflected the differences between the “VAS-Inf” and “VAD-Inf” groups defined earlier (Figure 1b), which would suggest the mechanisms underlying the compromised host resistance to *C. rodentium* that has been demonstrated in VAD mice [23,25]. 

### 2.6. WGCNA

For better comparability and consistency, the same data matrix in differential expression was used to build co-expression networks using the weighted correlation network analysis (WGCNA) package in R (version 3.4.4, Vienna, Austria) [54]. In other words, the dataset went through the same preprocessing steps (screening and normalization) as the differential expression analysis (Figure 1a). The application of the WGCNA tool is consistent with our earlier work, where the approaches were detailed [44]. To obtain distinct modules with moderate sizes, we set the minimum height for merging modules at 0.25 and the minimum module size to 30. According to the WGCNA standard usage, modules are henceforth referred to by their color labels, e.g., Colon(“Color”). Genes belonging to no specific modules were assigned to the Grey module. Heatmaps were depicted to visualize the expression pattern of each module using R package named pheatmap. Eigengenes are the first principal component of each set of module transcripts, describing most of the variance in the module gene expression [55]. As representations of individual modules, eigengenes were computed. Module–trait relationships were analyzed through correlating module eigengenes with the trait measurements, VA status, and infection status. The significance and correlation coefficient of each correlation between the traits and module eigengenes were visualized in a labeled heatmap, color-coded using red (positive correlation) and blue (negative correlation) shades. 

### 2.7. Functional Enrichment Analysis

Kyoto Encyclopedia of Genes and Genomes (KEGG) and Gene Ontology (GO) enrichment were assessed via clusterProfiler (v3.6.0) in R (version 3.4.4, Vienna, Austria) through one-tailed Fisher’s exact test, also known as the hypergeometric test [56]. The enrichment background was set as the filtered gene list of the colon (15,340 genes) or the SI (14,368 genes) (Figure 1a). As the statistical significance threshold for all functional enrichment analyses, padj (Benjamini and Hochberg adjusted for multiple comparisons) < 0.05 was used. 

## 3. Results

### 3.1. Model Validation: Infection Status and VA Status

Infection with *C. rodentium* was confirmed in all the mice used for transcriptomic analysis by measuring fecal shedding specific to this bacterium thrice weekly. In agreement with our previous reports [23,25], shedding increased after oral inoculation and plateaued during the period of peak infection (p.i. day 10). Bacterial counts increased with time (*p* < 0.0001) but did not differ between the infected VAD and VAS mice during either the 5-day course of the SI study (Appendix A) or the 10-day course of the colon study (Appendix A). Therefore, based on these kinetics and our previous kinetic studies of the passage of fluorescently tagged *C. rodentium* through the mouse intestinal tract [23,57], p.i. day 5 was selected for collection of the SI and p.i. day 10—for collection of the colon for further analysis.

The VA status of the mice was confirmed by measuring retinol concentrations in serum, which were significantly higher in the VAS mice than in their VAD counterparts (*p* < 0.001 [44]). These findings for both fecal shedding curves and serum retinol concentrations validated that our animal model is appropriate for the transcriptomic analysis that followed.

### 3.2. Mapping and Overview of the Differential Expression Results in the SI and the Colon

The flow of analysis and mapping results are illustrated in Figure 1a. Mapping revealed 24,421 genes, which were subsequently filtered for low-expression genes and normalized using DESeq2. Among the 15,340 genes expressed in the colon, the number of DEGs (Figure 1b) was as follows: 4524 genes corresponding to the infection effect under the VAS status (VAS-Inf, list I in the Appendix A); 4278 genes corresponding to the infection effect under the VAD status (VAD-Inf, list II in the Appendix A); 959 genes corresponding to the VA effect under infection (list III in the Appendix A); and 1329 genes corresponding to the interaction effect (list IV in the Appendix A). Since no DEG was discovered for the VAS-Inf or interaction effect in the SI, the results in the colon will be the primary focus of this paper.

### 3.3. Functional Categories of Colon DEGs in the VAS Infected vs. Non-Infected States (VAS-Inf Effect)

Of the 4524 DEGs corresponding to the VAS-Inf effect in the colon (list I in the Appendix A), 1424 were upregulated while 3100 were downregulated (Appendix A showing the volcano plot, the heat map, and enrichment categories). The genes belonging to the immune-related and epithelial cell (EC)-related categories are listed alphabetically by gene symbol in Appendix A, grouped first by the genes upregulated in VAS-Inf, followed by the genes downregulated in VAS-Inf, each compared to VAS alone, and subdivided into the genes likely originating in immune cells and the genes likely from ECs: the first included caspases, chemokine ligands, interleukin receptors, matrix metallopeptidases, antimicrobial proteins, and tumor necrosis factor-related genes, while those related to ECs included villin, actin, claudins, ROS production, endoplasmic reticulum (ER) stress signals, and goblet cell-specific transcripts. Some examples of upregulated DEGs that are also in the category of the top 30 most abundant genes are as follows: actin beta (Actb), mucin 2 (Muc2), chloride channel accessory 3B (Clca3b), Clca4b, and anterior gradient 2 (Agr2). Functional enrichment analyses for the 1424 upregulated DEGs in the colon in the VAS-Inf conditions suggested significant enrichment in the categories primarily relevant to inflammation, host–pathogen interactions, innate and adaptive immune functions (Appendix A), as well as cell proliferation, apoptosis, ER stress, and ROS production (Table 1). Several enrichment categories for biological pathways and processes previously shown to be induced by the *C. rodentium* infection were confirmed in the colon study (Table 1), adding to the validity of the model. 

Downregulated DEGs (Appendix A) contained fewer genes that represent immune signals while including several categories that are detailed further in the Discussion, including the genes involved in RA metabolic processes (Table 2); the genes involved in transcellular calcium absorption (Table 3); and the genes for the transporters that affect the absorption of water, Cl^−^, and Na^+^ (Table 4). Some most abundant downregulated DEGs were carbonic anhydrase 4 (Car4), carbonic anhydrase 2 (Car2), solute carrier family 26 member 3 (Slc26a3), and vitamin D (1,25-dihydroxyvitamin D3) receptor (Vdr). The GO and KEGG enrichment analyses for the 3100 DEGs that were downregulated during infection in the VAS colon revealed significantly enriched categories related to ion transport (Figure 2) as well as neurological, muscular, vessel, and developmental regulation (Appendix A, pink box on the left).

### 3.4. The Effect of Infection under VA Deficiency (VAD-Inf) and the Effect of the VA Status under Infection

We identified 4278 DEGs corresponding to the effect of infection under the VA deficiency conditions (VAD-Inf, list II in the Appendix A), which were visualized (Appendix A), among which 984 upregulated DEGs were enriched for functions such as defense response to the bacterium, IL-17 signaling pathway, acute-phase response, keratinization, water homeostasis, and neutrophil apoptosis (Appendix A, blue box on the right). The 3294 downregulated DEGs were enriched for categories like ion channel activity, muscle contraction, enteric nervous system, chemotaxis, solute sodium symporter activity, leukocyte activation, and Th17 cell differentiation (Appendix A, blue box on the right). The enriched functional categories above were further compared with those derived from the VAS-Inf comparison (left pink boxes of Appendix A). Whereas several GO and KEGG enrichment terms were up- or downregulated under both VAS and VAD status under infection (Appendix A, middle overlapping section in purple), it is clear that the VAS-Inf comparison, as compared to the VAD-Inf comparison, contained more biological functions in the upregulated category (Appendix A) and fewer in the downregulated category (Appendix A). These results showed that the *C. rodentium* infection interacted differentially with the VA status to alter the colon gene expression. The effect of the VA status under infection, reflected by 959 DEGs, was also identified (list III in the Appendix A) and visualized (Appendix A).

### 3.5. GO and KEGG Enrichment of DEGs: Four Interaction Scenarios Comparing Up- and Downregulated Gene Patterns in the Colon of VAS-Infected and VAD-Infected Mice

To further compare the gene expression patterns that distinguished the VAS-Inf group from the VAD-Inf group, we constructed four interaction “scenarios,” i.e., four subgroups of DEGs that were distinct based on their expression patterns, for the reason that these patterns may provide clues to mechanisms of interaction between the nutritional status and infection. DEGs corresponding to each of the four interaction scenarios in the colon are visualized in Figure 3 and listed in the Appendix A. Scenario 1 comprised the genes that were upregulated in the VAS-Inf comparison while being lower (six DEGs) or remaining unchanged (241 DEGs) in the VAD-Inf comparison; the higher expression of these genes was expected to be positively associated with host resistance and/or pathogen clearance in the VAS mice, while an absence of a response might suggest a reason why VAD hosts were less likely to effectively clear the pathogen and survive. Scenario 1 included 247 DEGs, with functional enrichment categories related to MHC class I, T cells, interferons, mitotic spindle, cytokine signaling, protein catabolism, apoptosis, and transcriptional regulation (Appendix A). Scenario 2 comprised the genes that were downregulated in the VAS-Inf comparison while being either higher (six DEGs) or unchanged (110 DEGs) in the VAD-Inf comparison; these transcripts may help to explain the greater severity of infection and a higher rate of mortality in the VAD mice. Scenario 2 included 116 DEGs (example genes see in Figure 4), but was, however, enriched for only two functional categories: “apical plasma membrane” and “apical part of cell,” both of which are cellular component categories in the GO database and likely related to EC functions. Scenario 3 comprised the genes that were unchanged in the VAS-Inf comparison but upregulated in the VAD-Inf comparison; however, although there were 90 DEGs in Scenario 3 (example genes see in Appendix A), there was no enrichment in any functional category. Finally, Scenario 4 comprised the genes that were unchanged in the VAS-Inf comparison while being downregulated in the VAD-Inf comparison; these genes might be likely to be associated with host resistance and pathogen clearance, i.e., the downregulation of those genes under VAD conditions could be a reason for the observed higher rate of lethality. Scenario 4 included 134 DEGs and was enriched for functional categories relevant to immune functions such as antigen processing and presentation, natural killer cell-mediated immunity, as well as cellular processes of proliferation, differentiation, migration, activation of T cells, and leukocytes (Figure 5).

### 3.6. Identifying WGCNA Modules That Were Significantly Correlated with Infection Effects and the Interaction of the VA Status and Infection in the Colon

The construction of co-expression gene networks via the WGCNA provides an independent systems biology perspective (Figure 1a). For this analysis, we set a soft threshold power β = 26; this identified 13 different modules in the colon. To determine if any of the modules of the co-expressed colon genes were correlated with the infection effect and the interaction effect, we tested the correlations between the traits (VA status, and infection status) and module eigengenes. Eigengenes are the first principal component of the given modules and may be considered as a representative of the gene expression profiles of the module. Six modules (Colon(Brown, Blue, Turquoise, Yellow, Purple, and Green–yellow)) were found to be significantly correlated with the infection status (*p*-values < 0.05, Figure 6a), among which the Colon (Blue, Yellow, Purple, and Green–yellow) modules were negatively correlated with the trait “infection status” (Figure 6a). Among those four modules, the Colon(Blue) module was the most significant (*p*-values = 9 × 10^−12^, correlation coefficient = −0.98). The correlation coefficient was negative, meaning the module contained genes with overall lower expression levels in the infected colons (Figure 6a). Not surprisingly, among the 5674 module members, more than half of the genes (*n* = 2967) were identified as the downregulated DEGs. The Colon(Blue) module mainly exhibited functional enrichment in neurological functions, transport, and extracellular matrix (Appendix A), meaning those activities were reduced in the colon of infected mice, comparing with their non-infected counterparts. The Colon(Turquoise) module was positively correlated with the infection status and enriched for functional categories involved in mRNA processing, chromosome segregation, protein catabolic process, etc. (Appendix A). The Colon(Brown) and Colon(Purple) modules were simultaneously associated with VA and the infection status (Figure 6b,c). Colon(Purple) exhibited enrichment of the “anion transmembrane transporter activity,” whereas Colon(Brown) was not enriched for any functional categories (Appendix A). For each module, a module member gene list (Appendix A) and an eigengene bar graph (Appendix A) are provided.

## 4. Discussion

### 4.1. Approach and Validation

In this study, we used transcriptional profiling and bioinformatics in an unbiased, exploratory manner to better understand the ways in which the VA status and gut infection modulate the expression of genes in the SI and the colon, which may in turn suggest mechanisms of host defense and thus help to explain the observations that VA-deficient hosts are more likely than their VA-adequate counterparts to die from the *C. rodentium* infection [23,25]. An admitted limitation of the RNAseq analysis is that it only provides information on transcript levels, while other stages of regulation, such as protein expression and activity, are not detected. Nevertheless, RNAseq provides a comprehensive analysis that can be subjected to further mining using bioinformatics approaches. Moreover, RNAseq also does not rely on individual reference genes, an advantage over traditional PCR methods. Because there has been no previous comprehensive analysis of the gene expression patterns comparing the VAS and VAD nutritional conditions under both non-infected and infected states, the current study was designed to address these gaps. Firstly, we found that infection affected gene expression in the colon, but not the SI (which, however, was affected by the VA status, see [44]), and, therefore, our discussion herein is focused on the colon. As was illustrated in Figure 1a, the design used in this study allowed for several comparisons: the effect of the VA status (VAS vs. VAD) in the absence of infection, as previously reported [44], and the effects of infection in the VAS condition (VAS-Inf), which revealed several interesting patterns of DEGs that concern *(i)* VA metabolic genes, *(ii)* calcium transport and VD-related genes, and *(iii)* ion and water transport, and which, therefore, may be of interest in the relationship to diarrheal disease. We then discuss the main differences observed when comparing infection in the VAS and VAD conditions (interaction effect, which we divided into four scenarios). Finally, we note that the complete gene lists are provided as the Appendix A, and the datasets are available at NCBI GEO (accession No. GSE143290), providing additional information for further analysis.

We first validated our model by measuring the serum retinol, which was higher in the VAS animals than in the VAD animals [44], and by monitoring fecal shedding, as in previous studies, to ensure the infection status (Appendix A). As we began our RNAseq analysis, we looked for the DEGs that are known as *C. rodentium*-responsive in the colon to assure replication under our VAS-Inf conditions. Such genes included processes of epithelial hyperplasia and apoptosis [58], goblet cell depletion, as well as innate and adaptive immune responses [6], which were replicated in the present dataset (Table 1 and Appendix A). This is in line with the fact that Colon(Turquoise), a WGCNA module positively correlated with the infection status, was enriched for biological functions such as chromosome segregation, and protein catabolic process (Appendix A). With respect to epithelial hyperplasia and apoptosis, a hallmark pathological feature of the *C. rodentium* infection is the transmissible murine crypt hyperplasia (TMCH) [58], defined by the thickening of colonic mucosa and caused by the excessive induction of epithelial repair and regeneration mechanisms. During peak infection, the expedited colonic hyperplasia and the apoptosis (Table 1) can altogether serve protective roles for the host against *C. rodentium* because both processes can speed up the life cycle of the ECs, a proportion of which may have been attached by *C. rodentium*. When the rate of differentiation, migration, detachment, and programmed cell death can exceed that of the proliferation of the attached *C. rodentium*, gradual clearance of the pathogen will take place. Regarding goblet cell depletion, our data summarized in Table 1 agree with a previously published work that goblet cell activities are induced by and are protective against A/E pathogens, as increased mucin mRNA expression and mucus thickness during the *C. rodentium* infection were observed [18]. Mucin can displace the pathogen to the outer mucus layer and aid in flushing away the pathogen [73]. Some genes such as Il17a, Il22, and Ifng that would be expected to increase were not detected as DEGs in our study [74,75], perhaps due to the filtering and the expression-level criteria used. However, inspection of the raw data prior to filtering showed that there were changes in the expected direction in these genes. Moreover, the more abundant expressions of the receptors of those cytokines (Il-17 receptors a–e, IL-22 receptors a1 and a2, IFNg receptors Ifngr1 and Ifngr2), combined with other DEGs, helped to drive the enrichment categories “IL-17 signaling pathway,” “Th17 cell differentiation,” and “response to interferon-gamma,” which can indirectly serve as a proof of the principle that cytokines were induced during peak infection.

### 4.2. DEGs Positively Associated with Host Resistance under the VAS Status

The analysis of DEGs expressed under VAS-Inf conditions also revealed upregulated genes that are likely attributable to immune cells in the lamina propria, such as genes for caspases, chemokine ligands, interleukin receptors, matrix metallopeptidases, antimicrobial proteins and tumor necrosis factor-related genes, etc., as well as genes likely derived from ECs, including villin, actin, claudins, ROS production, ER stress signals, and goblet cell-specific transcripts (Appendix A), and the processes related to inflammation, host–pathogen interactions, innate and adaptive immune functions were increased (Appendix A), as well as cell proliferation, apoptosis, ER stress, and ROS production (Appendix A). 

The host response to *C. rodentium* depends on the VA status, wherein the VAD mice showed a lower survival rate and a slower clearance rate [23,25]. To study the key pathways mediating the mitigated symptoms in the VAS hosts, the interaction effect was analyzed. Because the transcriptomic interaction between the VA status and the infection response could be either positive or negative, the analysis resulted in four different scenarios, of which Scenarios 1 and 4 were positively associated with a more robust host resistance. Scenario 1 contained the DEGs that were higher during infection in the VAS condition while being lower or unchanged with infection in the VAD condition. Therefore, this group of genes might be essential for the host response against the *C. rodentium* infection, and, conversely, the absence of their response might explain why the VAD hosts were unable to effectively clear the pathogen [23,25]. The Muc4 DEG belonged to Scenario 1. Muc4 mRNA was not influenced by the VA status (VAS vs. VAD) in a naïve mouse colon [44]. Herein, Muc4 was significantly upregulated during peak *C. rodentium* infection only in the VAS mice (VAS-Inf), but not in the VAD mice (VAD-Inf), suggesting it might play a protective role against the *C. rodentium* infection. The expression pattern of Muc4 (Appendix A) is not only a good example as the interaction effect, but also suggests that VA signaling in the colon is a prerequisite for the induction of Muc4 during colitis. 

*C. rodentium*-infected mice have been widely used to investigate innate and adaptive mucosal immunity. Lymphocytes, especially B cells and CD4^+^ T cells, are crucial for the resistance to the *C. rodentium* infection [66,67]. The cytokine IL-17 is required for the clearance of the pathogen [24,26,70,71]. Th17 cells are a distinct branch of mucosal immunity that deals with A/E lesions, and therefore can be induced by the *C. rodentium* infection [76]. Atarashi et al. [12] demonstrated that EC adhesion by *C. rodentium* and the subsequent ROS production mediated the induction of Th17 cells. In the current RNAseq dataset, upregulated DEGs nitric oxide synthase 2 (Nos2), dual oxidase 2 (Duox2), and dual oxidase maturation factor 2 (Duoxa2) (Table 1 and Appendix A), together with other DEGs, drove the enrichment of the “reactive oxygen species metabolic process” (Table 1 and Appendix A). Furthermore, “Th17 cell differentiation” was a KEGG enrichment term for Scenario 4 of the interaction effect, that is, for the genes that were unchanged in VAS-Inf while being downregulated in VAD-Inf and which, therefore, might likely be associated with impaired host resistance and pathogen clearance under VAD conditions (Figure 5a and Table 1).

Infection by several enteropathogens, including *C. rodentium*, leads to a drastic reduction in the number of the phenotypically distinct goblet cells, a process defined as the “goblet cell depletion” [58]. This may seemingly contradict the increased transcriptional activities in goblet cells during the *C. rodentium* infection (Muc, Retnlb, Clcas, and Fut2) observed in our study (Appendix A), but a closer examination indicates that this evidence may indeed line up well with each other. Goblet cell depletion, reflected by a reduction of periodic acid–Schiff (PAS) staining on histological colon sections, was actually caused by a reduction of the mucin glycoprotein content of goblet cells rather than an actual lineage loss of goblet cells. 

Whereas the majority of previous research was focused on the genes induced by *C. rodentium*, the downregulated DEGs were more than double the number of the upregulated DEG under the VAS-Inf conditions (3100 vs. 1424, Appendix A and list I of the Appendix A); those downregulated DEGs included categories of genes that may be most closely associated with the nutritional status and/or with diarrheal disease, as discussed next.

### 4.3. Three Categories of Downregulated DEGs Related to Nutritient Utilization and Ion Transport Relevant to Diarrheal Disease

Three groups of genes were identified from this study that could be of particular importance for understanding nutrient processes during infection and infection-related diarrhea. Firstly, our results showed that the *C. rodentium* infection concordantly downregulated VA metabolic genes in the VAS colon. Several genes involved in RA metabolic processes, which are known to be inducible by RA, were significantly downregulated in the colon of the VAS mice during peak infection (Table 2), including Aldh1a1, Aldh1a2, Cyp26b1, and Lrat [77,78]. Enzymes RALDH1 and RALDH2 (encoded by Aldh1a1 and Aldh1a2, respectively) convert retinal to RA [78], whereas cytochrome P450 enzymes (CYP26), present in a variety of tissues, catabolize RA to its less bioactive form and thus may be critical for preventing the buildup of high concentrations of RA intracellularly [79]. Similarly, lecithin retinol acyltransferase (LRAT) is also considered among the most important genes for regulating VA metabolism by esterifying retinol to its storage form, retinol ester. Previously, the expression levels of several of these genes were identified as being reduced in VAD vs. VAS SI [44]; however, here, differences were observed due to infection in the VAS host. Besides the reduction of these RA-responsive genes, a uniform downregulation was observed for other VA-metabolic genes (Bco2 (a beta-carotene cleavage enzyme that provides a precursor for RA synthesis [80]); Rarb, Rxrg (nuclear receptors mediating RA signaling), Rbp2 (binds retinol for its intracellular transport and conversion), Adh1, Adh7, Rdh5 (enzymes that catalyze the conversion from retinol to retinal), and Crabp1 (an intracellular RA-binding protein [81]), as well as upregulation of the two rate-limiting enzymes (Rdh1 and Dhrs9) for the production of retinal as a precursor of RA [82] (Table 2). These changes suggest that infection reduces the capacity to produce RA, while it may also interfere with VA storage and RA catabolism. Overall, the pattern of changes in the expression of genes suggests a reduced ability to utilize VA and/or produce RA for retinoid signaling. Although on a systemic level, the vast majority of VA absorption occurs higher up in the SI, a small amount may be directly absorbed by the colon, considering the recent findings that water-soluble vitamins, known to be uptaken in bulk in the SI, now found efficiently absorbed in colon [83]. Rodent colons do contain VA [84], and the colonic gene expression profile is influenced by the VA status [44], indicating there is either direct absorption in the colon and/or some VA is absorbed in SI then allocated to colon via blood circulation. Since VA needs to be converted to its bioactive form to take effect, the reduced local expression of RA metabolic genes in colon can compromise the health of colonocytes, as well as the education and differentiation of colonic immune cells (e.g., DCs, lymphocytes, and ILCs) [43]. Not necessarily expected, the *C. rodentium* infection in the colon recapitulated some of the effects previously noted for VA deficiency in the SI [44], suggesting there could be regulated VA metabolism, even in the colon, under the control of the VA status [44] or bacterial infection. 

Secondly, we also observed that the *C. rodentium* infection concordantly downregulated the VD metabolic and calcium transport-related genes in the VAS colon. An interesting finding was the almost uniform downregulation of Vdr and the genes involved in calcium absorption during infection in the VAS-Inf group compared to the VAS control mice (Table 3). VDR is a nuclear receptor that induces the expressions of several genes in the transcellular calcium absorption process. We observed reductions during peak infection in Cacna1d (which encodes Ca_v_1.3, a calcium channel allowing for Ca^2+^ entry through the apical membrane of enterocytes), S100g (which encodes calbindin CB_9k_, a binding protein that chaperones Ca^2+^ to move from apical to basolateral membrane in enterocytes), Atp2b2, Atp2b3, Atp2b4 (which encode the plasma membrane Ca^2+^ ATPase family that extrude Ca^2+^ into the blood stream across the basolateral membrane), and Slc8a1 (which encodes sodium calcium exchanger, or NCX1, which also plays a role in the extrusion step) (Table 3) [85]. From these results, it appears that infection causes an apparently coordinate reduction in processes necessary for calcium uptake. 

In addition to being a regulator of the calcium absorption process, VD is also known as a regulator of mucosal immunity in the gut [86]. VD plays dual roles on activating/suppressing cells in the mucosal immune system, which seem to be dependent on the phase of the infection, where at the beginning of the infection VD is required for the proper initiation of the immune response, while the immunosuppressive role of VD emerges during the peak of infection and becomes essential during the resolution phase of the infection [87]. Macrophages, DCs, T cells, and B cells are all VD targets since they all express VDR [87]. Immune cells generally take 2–3 days after infection to become activated and maximize their VDR expression [87]. During the early phase of the *C. rodentium* infection, VD is essential for the differentiation of ILC3 and the production of IL-22, which may be required for the normal expansion of Th17 cells during the later phase of infection [88]. Previous studies using VD-deficient mice showed some similar changes, such as significantly fewer ILC3 cells and lower IL-22 production, failure to expand the Th17 population, clearance of the pathogens at a lower rate, more severe infection, and more rapid mortality comparing with their VD-sufficient counterparts [88]. Those results resemble our observation during VAS-Inf, when Vdr and the vast majority of VD target genes were downregulated (Table 3); meanwhile, enrichment terms for “interleukin-6 production,” “TNF signaling pathway,” “response to interferon-gamma,” and “IL-17 signaling pathway” were increased (Appendix A). To sum up, these results suggest the idea that the *C. rodentium* infection may result in an overall reduction of the effective concentration of calcitriol in the colon, which may aid the host resistance/survival mechanisms through partially releasing the immunosuppressive effect of VD. Additionally, considering that IL-17 is required for the clearance of this A/E pathogen [24,26,70,71], reduced VA and vitamin D signaling locally in the colon under the state of immune activation status during peak *C. rodentium* infection might have been utilized by the host (either passively or actively) as an “on-switch” for the Th17 immune response. However, further experiments are necessary to test this possible connection. 

Thirdly, our study provides evidence that *C. rodentium* infection concordantly downregulated ion transport pathways, which may underlie *C. rodentium*-induced diarrhea. Water loss during diarrhea is a major cause of mortality in experimental models and in humans [14,29,89,90]. Although the secretion of fluid and mucus are important mucosal defense mechanisms that can dilute and wash away pathogens and their products from the epithelial surface, diarrhea can be fatal due to the fluid losses, hypovolemia, and organ failure [91]. In the current study, downregulation of genes involved in Cl^−^ absorption (Slc26a3), Cl^−^ secretion (Cftr and Ano1), Na^+^ absorption (Slc9a2 and Chp2), and HCO_3_^−^ homeostasis (Car 2, 3, 4, 11, 14, 15 and Best2) were observed in VAS-Inf (Table 4), resulting in enriched functional categories of “sodium ion transport,” “regulation of cytosolic calcium ion concentration,” “regulation of cation transmembrane transport,” “potassium ion transport,” and “regulation of ion transmembrane transport” (Figure 2). In addition, Colon (Blue) and Colon (Purple) were two WGCNA modules suggesting lower transmembrane transport activities in infected colon (Appendix A, Figure 4c, and Appendix A).

Regarding chloride transepithelial transport, DEGs were identified for both absorption and secretion-related processes. Slc26a3 encodes for the Downregulated in Adenoma (DRA), a Cl^−^/HCO_3_^−^ exchanger, which mediates the apical chloride absorption into the ECs and the secretion of bicarbonate into the lumen. Mutation of DRA is associated with an intestinal disorder characterized by watery diarrhea, severe dehydration, high levels of fecal chloride, hypochloremia, and hyponatremia [92]. In previous studies, the downregulation of Slc26a3 have been accused for the fatal diarrhea, characterized by the loss of water in the stool [28]. The reduction of the Slc26a3 transcription may be resulted from the elevated TNFα signaling during peak *C. rodentium* infection [93]. Regarding chloride secretion, Cftr and Ano1, two transporters located on the apical membrane of colonic ECs that secrete chloride into the lumen [94] were found drastically reduced during VAS-Inf (Table 4), suggesting downregulation of Cl^−^ loss through feces in VAS mice during diarrhea. This may have compensated the reduction of Cl^−^ absorption, which may be essential for the survival of the VAS mice. Interestingly, Cftr and Ano1 were two genes belonging to Scenario 2 of the Interaction effect, which were not effectively downregulated in VAD-Inf (Figure 4a,c). This may suggest that excessive Cl^−^ secretion into the lumen may have exacerbated the water loss through diarrhea.

With respect to sodium absorption, previous studies have showed suppression of Slc9a2, Slc9a3, and Chp2 expression by *C. rodentium* infection [28,29], wherein the reduction of Slc9a2 and Chp2 were replicated in the VAS-Inf comparison of the current study (Table 4). NHE2 (encoded by Slc9a2) and NHE3 (encoded by Slc9a3) are both major Na^+^/H^+^ exchangers responsible for the apical sodium absorption, pH regulation, and fluid balance in the intestine [95,96]. Calcineurin-like EF hand protein 2 (CHP2), an essential cofactor for the NHE family members, supports the Na^+^/H^+^ exchange activity [97]. The differential expression of these genes may also contribute to a disturbed ion balance. 

With respect to bicarbonate metabolism, carbonic anhydrases (CAs) catalyze the reversible dehydration/hydration of CO_2_ and water [98], thus maintaining the bicarbonate pool required for the ion-exchange activity of chloride and bicarbonate (e.g., DRA) and supplying protons for the apical membrane ion exchangers (e.g., NHEs and cHKA), and therefore, inhibition of carbonic anhydrases is associated with marked reductions in bicarbonate secretion as well as chloride, sodium, and water absorption [29,96,99]. Significant suppression of Car2 and Car4 by *C. rodentium* infection were observed previously [18,28,29] and in our study (Table 4). Car4 and Car14 encode for CA isoforms that are membrane-bound, whereas Car2, Car3, and Car13 encode cytosolic isoforms. Bestrophin-2 (Best2), a gene encoding a HCO_3_^–^ channel in the basolateral membrane, was also found to be downregulated by the *C. rodentium* infection in previous works [18] and in our study under the VAS-Inf condition (Table 4). However, as a DEG belonging to Scenario 2 of the Interaction effect, it was not significantly downregulated in the VAD-Inf condition (Figure 4b). Slc4a7, encoding a Na^+^-HCO_3_^−^-cotransporter named NBCn1 in the basolateral membrane of the goblet cells in the colonic crypts [100,101], were found significantly upregulated during VAD-Inf, but was not significantly changed during the VAS-Inf (Scenario 3 of Interaction effect, Appendix A). HCO_3_^−^ is of the utmost importance for the build-up of the mucus layer [102]. Since the upregulation of Slc4a7 was coincident with the increased mortality rate in the VAD mice, one possible explanation is that higher NBCn1 channel facilitated the HCO_3_^−^ transportation from the blood stream into the colonocytes, which supports a higher mucus production. Even though extra mucus can help isolate, dilute, and flush away the pathogens, it also requires more water to hydrate the mucin molecules, resulting in more water loss and less water absorption in the VAD hosts. Dehydration, rather than sepsis, is the major cause of lethality in the *C. rodentium* infection within the susceptible mouse strains [29].

Therefore, our study supports an overall downregulation of transcriptional machinery to maintain the intracellular HCO_3_^−^ level, and decreased chloride/sodium absorption during VAS-Inf, which is concordant with a mild diarrhea and high survival rate in the VAS mice. In contrast, the diminished compensatory effect reflected by the expression patterns of Cftr, Ano1, Best2, Slc4a7 and possibly other key genes in Scenario 2 and 3 may have exacerbated the water loss during the VAD-Inf, contributing to the higher mortality in the VAD hosts. 

## 5. Conclusions

The RNAseq approach used in this study provides an abundance of data for these and further analyses. Finding that the response of VAD mice to infection differed from that of VAS mice with similar infection, as shown in terms of heat maps and several gene enrichment pathways, helps to generate new hypotheses for future experiments. Given that VAS mice have been shown to survive *C. rodentium* infection whereas a significant proportion of VAD mice succumb [23,25], it is of interest that several genes related to chloride, sodium, and bicarbonate ions, which responded in the VAS-Inf condition, did not respond in the VAD-Inf condition (Scenario 2). These observations lend support to the idea that that the host’s response related to ion and water loss is crucial with respect to preventing mortality due to this gut infection, and that a deficiency of VA handicaps the host’s ability to make an appropriate ion transport and fluid response.

## Figures and Tables

**Figure 1 nutrients-14-01563-f001:**
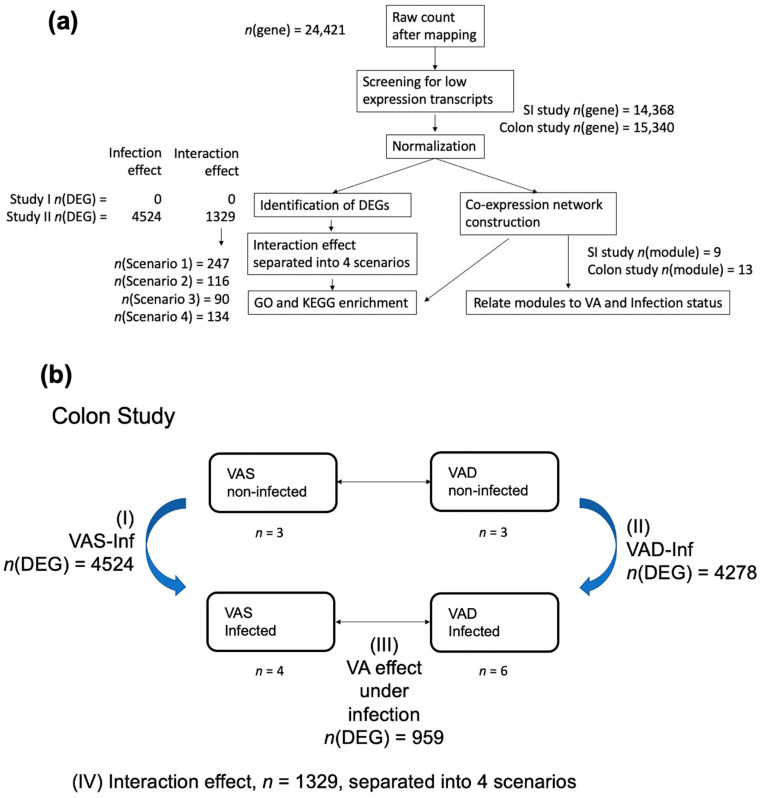
Overview: bioinformatics analyses results. (**a**) Initial mapping identified 24,421 genes. After the filtering of low-expression genes, 15,340 genes in the colon and 14,368 genes in the SI were retained, respectively. Among the 15,340 colon genes expressed in the colon, 4524 DEGs corresponding to the VAS-Inf effect and 1329 DEGs corresponding to the interaction effect were identified. However, no DEGs corresponding to the VAS-Inf or the interaction effect were found in the SI dataset. Due to the complexity of the interaction effect, different scenarios were isolated based on gene expression patterns, creating sublists of DEGs of the interaction effect, among which four of the scenarios were used in the enrichment analysis and visualization. Thirteen co-expression modules were identified in the colon via the WGCNA; (**b**) *n*(DEG) was the number of DEGs in each of the comparisons (padj < 0.05 and |FoldChange| > 2). The number of DEGs corresponding to the VAS-Inf effect (list I), the VAD-Inf effect (list II), the VA effect under infection (list III), and the interaction effect (list IV) was 4524, 4278, 959, and 1329, respectively. For the full length of each gene list see the Appendix A. **Abbreviations**: Infection effect under the VAS status (VAS-Inf), infection effect under the VAD status (VAD-Inf), differentially expressed gene (DEG), Gene Ontology (GO), weighted gene co-expression network analysis (WGCNA), Kyoto Encyclopedia of Genes and Genomes (KEGG), small intestine (SI).

**Figure 2 nutrients-14-01563-f002:**
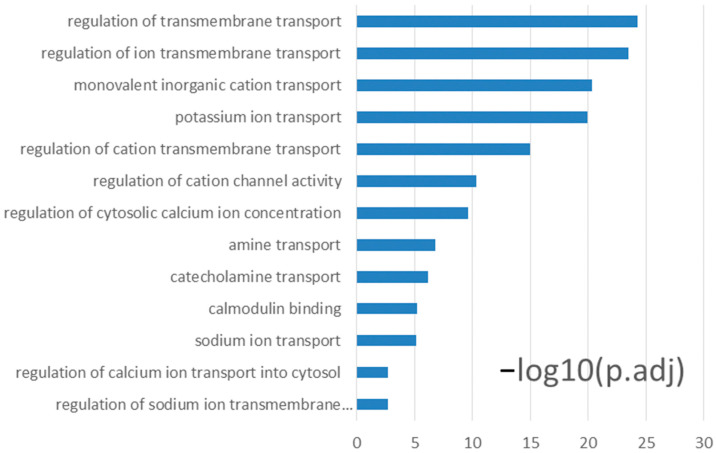
Transport and ion concentration-related GO and KEGG enrichment terms downregulated during VAS-Inf in the colon. VAS-Inf comparison of infected VAS (*n* = 4) versus non-infected VAS (*n* = 3). Functional enrichment analysis was performed on the 3100 downregulated DEGs derived from this comparison. Criterion: padj < 0.05. For a full list of the DEGs reflecting VAS-Inf comparison in the colon dataset see list I in the Appendix A. **Abbreviations**: infection effect under the VAS status (VAS-Inf), Kyoto Encyclopedia of Genes and Genomes (KEGG), differentially expressed gene (DEG), Gene Ontology (GO), padj (Benjamini and Hochberg-adjusted *p*-value, hypergeometric test).

**Figure 3 nutrients-14-01563-f003:**
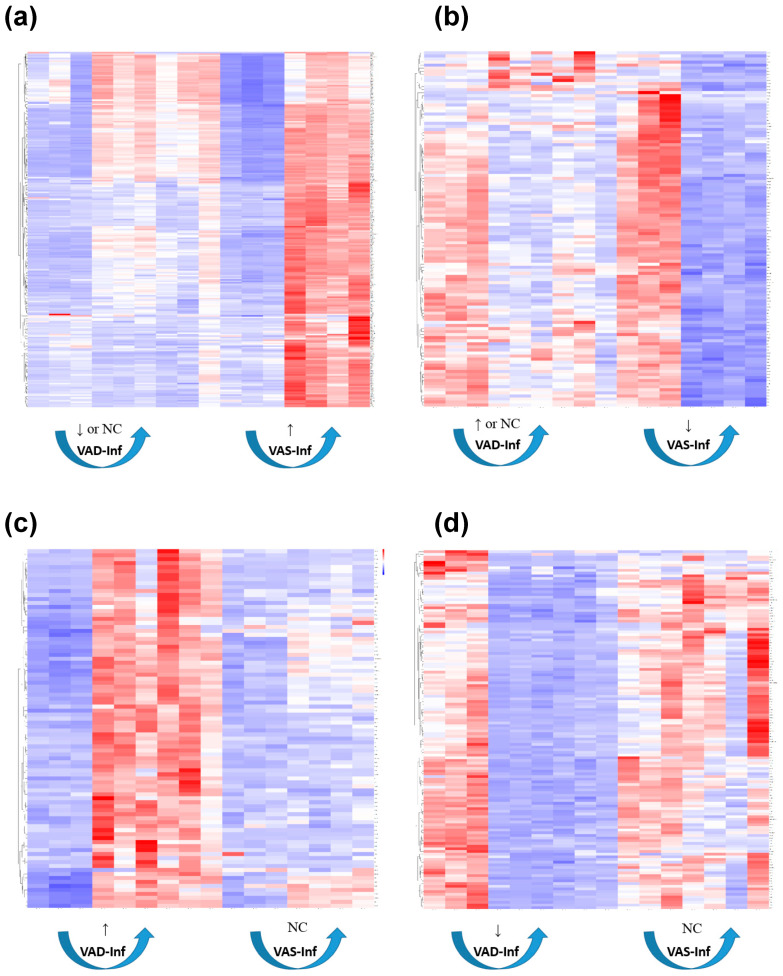
Heatmap for DEGs corresponding to four different scenarios for the interaction effect in the colon. (**a**) Scenario 1. (**b**) Scenario 2. (**c**) Scenario 3. (**d**) Scenario 4. For full lists of the genes corresponding to those scenarios see the Appendix A. Red: higher expression; blue: lower expression. Non-infected VAD: columns 1, 2, and 3. Infected VAD: columns 4, 5, 6, 7, 8, and 9. Non-infected VAS: columns 10, 11, and 12. Infected VAS: 13, 14, 15, and 16. ↑ means upregulation, ↓ means downregulation. **Abbreviations**: differentially expressed gene (DEG), no change (NC), vitamin A-sufficient (VAS), vitamin A-deficient (VAD), infection effect under the VAS status (VAS-Inf), infection effect under the VAD status (VAD-Inf).

**Figure 4 nutrients-14-01563-f004:**
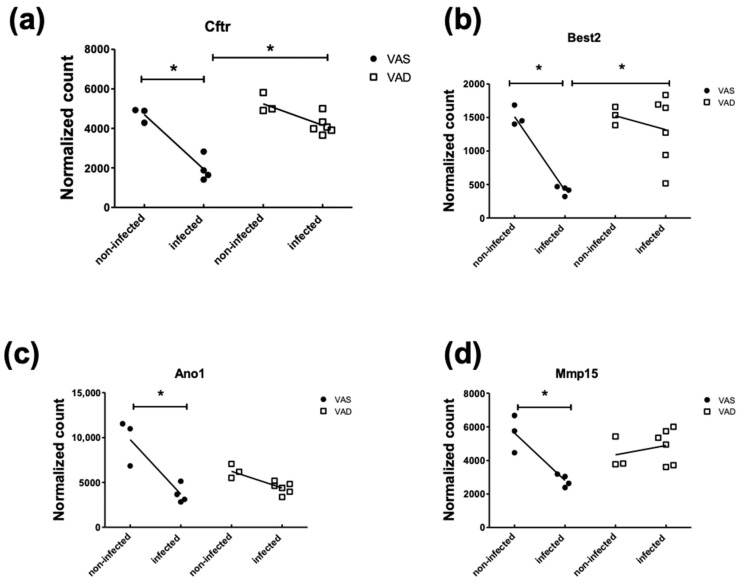
Representative DEGs corresponding to Scenario 2 in the colon. Expression of genes (**a**) cystic fibrosis transmembrane conductance regulator (Cftr), (**b**) Bestrophin-2 (Best2), (**c**) anoctamin 1 calcium activated chloride channel (Ano1), and (**d**) Matrix metalloproteinase 15 (Mmp15) in all four conditions. Note: * in the differential expression analysis, |FoldChange| > 2 and adjusted *p*-value < 0.05. For a full list of the genes corresponding to Scenario 2 of the interaction effect, see Scenario 2 in the Appendix A. **Abbreviations**: differentially expressed gene (DEG), vitamin A-sufficient (VAS), vitamin A-deficient (VAD).

**Figure 5 nutrients-14-01563-f005:**
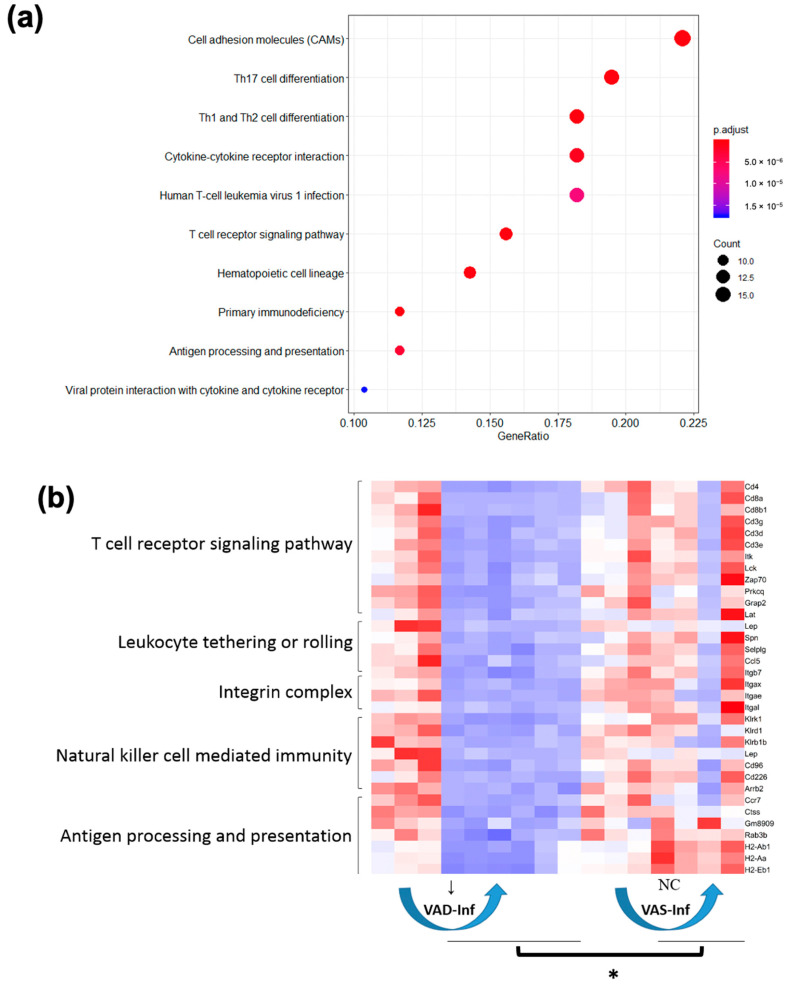
Biological functions altered in Scenario 4 in the colon. (**a**) Top 10 KEGG terms enriched for Scenario 4. Criteria: padj < 0.05. Count: number of the Scenario 4 genes that fell in the corresponding category. GeneRatio: count/(total number of genes in the test). (**b**) Heatmap of the DEGs enriched for representative GO and KEGG terms. Red: higher expression; blue: lower expression. Non-infected VAD: columns 1, 2, and 3. Infected VAD: columns 4, 5, 6, 7, 8, and 9. Non-infected VAS: columns 10, 11, and 12. Infected VAS: 13, 14, 15, and 16. ↓ means downregulation. Note: * expression levels of those genes in the VAS infected group were significantly higher than in the VAD infected group (adjusted *p*-value < 0.05 and |FoldChange| > 2). For a full list of genes corresponding to Scenario 4 of the interaction effect see Scenario 4 in the Appendix A. **Abbreviations**: Kyoto Encyclopedia of Genes and Genomes (KEGG), Gene Ontology (GO), padj (Benjamini and Hochberg-adjusted *p*-value, hypergeometric test), differentially expressed gene (DEG), no change (NC), vitamin A-sufficient (VAS), vitamin A-deficient (VAD), infection effect under the VAS status (VAS-Inf), infection effect under the VAD status (VAD-Inf).

**Figure 6 nutrients-14-01563-f006:**
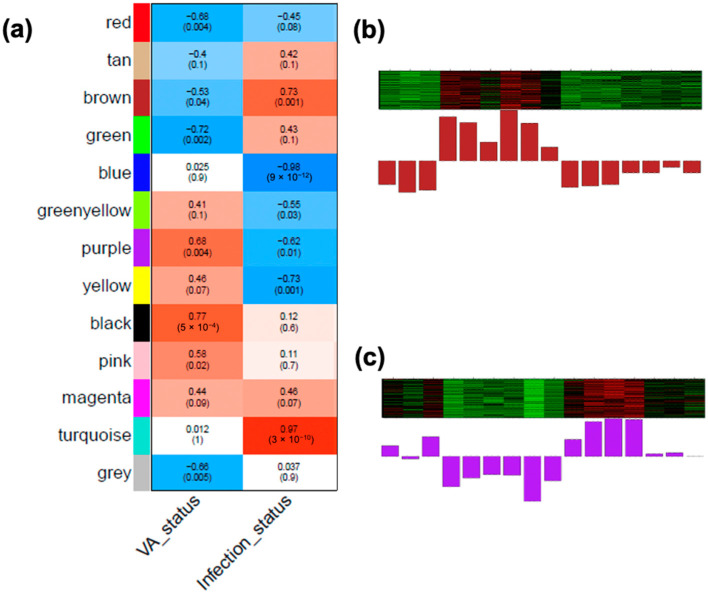
Correlation of modules with VA and the infection status in the colon. (**a**) Correlation of biological traits with colonic module eigengenes, calculated by the WGCNA. Upper numbers in each cell: correlation coefficient between the trait (VA and the infection status) and the module eigengene. Lower numbers in parentheses: *p*-value of the corresponding correlation. Significant correlation with the infection status were obtained on the Colon(Blue), Colon(Turquoise), Colon(Brown), Colon(Yellow), Colon(Purple), and Colon(Green–yellow) modules, among which Colon(Brown) and Colon(Purple) were significantly correlated additionally with the VA status. The eigengene values (Y-axis) of Colon(Brown) and Colon(Purple) across samples (X-axis) are visualized in (**b**,**c**). **Abbreviations**: vitamin A (VA), weighted gene co-expression network analysis (WGCNA).

**Table 1 nutrients-14-01563-t001:** Enrichment categories for the biological pathways and processes induced by the *C. rodentium* infection confirmed in the colon study.

Biological Process	Ref.	Evidence, VAS-Inf	Evidence Scenario 1	Evidence Scenario 4
Epithelial hyperplasia	[58]	“mitotic nuclear division”, “DNA replication”, and “cholesterol biosynthetic process”	“mitotic spindle organization”	
Apoptosis	[13,14,58]	“activation of cysteine-type endopeptidase activity involved in apoptotic process”	caspases 1, 3, 4, and 7, “cysteine-type endopeptidase activity involved in apoptotic process”	
Goblet cell depletion	[6,18]	Muc 1, 2, and 4, chloride channel accessory family that are products of goblet cells (Clca3a1, Clca3a2, Clca3b, Clca4a, and Clca4b), and enzymes that may affect the glycosylation of mucins (Glt28d2 [59] and Fut2 [60,61,62]	Muc4, Agr2	
Innate immune response	[63,64,65]	“neutrophil chemotaxis,” “monocyte chemotaxis,” and “regulation of leukocyte migration”	Tlr2, Myd88, Cxcl9, and Ccl8	“natural killer cell mediated immunity,” “leukocyte tethering or rolling,” and “antigen processing and presentation”
T cell response	[66,67,68,69]	“adaptive immune response,” “lymphocyte mediated immunity,” “response to interferon-gamma,” and “T cell mediated cytotoxicity,”	“T cell mediated cytotoxicity,” “leukocyte mediated cytotoxicity,” “T cell mediated immunity,” and “leukocyte mediated immunity”	“T cell receptor signaling pathway,” and “integrin complex”
IL-17 response	[11,12,24,26,70,71,72]	nitric oxide synthase 2 (Nos2), dual oxidase 2 (Duox2), and dual oxidase maturation factor 2 (Duoxa2), Cebpd, Saa 2, 3, and 4; “reactive oxygen species metabolic process” and “IL-17 signaling pathway”		“Th17 cell differentiation”

**Abbreviation**: Infection effect under the VAS status (VAS-Inf).

**Table 2 nutrients-14-01563-t002:** VA metabolic genes altered by the *C. rodentium* infection in the colon study.

Upregulated during Infection					
	Base Mean	Fold Change	padj	ENTREZID	Full Name
Akr1c18	54.22	7.46	2.79 × 10^−6^	105,349	aldo–keto reductase family 1 member C18
Aldh1a3	1061.53	2.32	4.70 × 10^−4^	56,847	aldehyde dehydrogenase family 1 subfamily A3
Dhrs9	1262.32	17.44	2.60 × 10^−29^	241,452	dehydrogenase/reductase (SDR family) member 9
Rdh1	24.83	2.59	4.37 × 10^−4^	107,605	retinol dehydrogenase 1 (all trans)
**Downregulated during infection**					
Adh1	29,054.77	−2.19	8.36 × 10^−6^	11,522	alcohol dehydrogenase 1 (class I)
Adh7	14.73	−3.50	4.95 × 10^−3^	11,529	alcohol dehydrogenase 7 (class IV) mu or sigma polypeptide
Aldh1a1	505.92	−2.12	3.20 × 10^−4^	11,668	aldehyde dehydrogenase family 1 subfamily A1
Aldh1a2	114.85	−2.72	5.83 × 10^−3^	19,378	aldehyde dehydrogenase family 1 subfamily A2
Bco2	512.13	−2.27	4.92 × 10^−4^	170,752	beta-carotene oxygenase 2
Crabp1	13.86	−4.61	5.53 × 10^−5^	12,903	cellular retinoic acid-binding protein I
Cyp26b1	41.71	−5.57	2.94 × 10^−6^	232,174	cytochrome P450 family 26 subfamily b polypeptide 1
Lrat	42.31	−8.02	1.14 × 10^−8^	79,235	lecithin retinol acyltransferase (phosphatidylcholine-retinol O-acyltransferase)
Rarb	65.16	−3.31	1.63 × 10^−7^	218,772	retinoic acid receptor beta
Rbp2	237.32	−3.23	9.94 × 10^−8^	19,660	retinol-binding protein 2 cellular
Rdh5	120.00	−2.24	8.23 × 10^−9^	19,682	retinol dehydrogenase 5
Rxrg	18.21	−10.12	9.98 × 10^−9^	20,183	retinoid X receptor gamma
Stra6l	517.21	−3.29	1.76 × 10^−14^	74,152	STRA6-like

VAS-Inf comparison of infected VAS (*n* = 4) versus non-infected VAS (*n* = 3). The comparison included 4524 DEGs, of which 1424 were upregulated, 3100 downregulated, and 10,816 showed no change. The list order is ranked alphabetically by the official gene symbol. Only the DEGs related to VA metabolism are included here. For a full list of DEGs reflecting the VAS-Inf comparison in the colon dataset see list I in the Appendix A. **Abbreviations**: vitamin A-sufficient (VAS); infection effect under the VAS status (VAS-Inf); differentially expressed gene (DEG); padj: DESeq2-computed adjusted *p*-value; base mean: mean expression level across all the 16 libraries, with DESeq2 normalization.

**Table 3 nutrients-14-01563-t003:** VDR-related gene expression altered by the *C. rodentium* infection in the colon.

Upregulated during Infection					
	Base Mean	Fold Change	padj	ENTREZID	Full Name
Trpv6	485.72	5.55	4.31 × 10^−6^	64,177	transient receptor potential cation channel subfamily V member 6
**Downregulated during Infection**					
Atp2b2	35.55	−3.80	1.37 × 10^−5^	11,941	ATPase Ca^++^-transporting plasma membrane 2
Atp2b3	68.79	−3.68	6.71 × 10^−7^	320,707	ATPase Ca^++^-transporting plasma membrane 3
Atp2b4	3441.90	−4.27	1.68 × 10^−12^	381,290	ATPase Ca^++^-transporting plasma membrane 4
Cacna1d	227.14	−3.34	2.02 × 10^−9^	12,289	calcium channel voltage-dependent L-type alpha 1D subunit
Cyp27a1	889.90	−5.36	4.23 × 10^−15^	104,086	cytochrome P450 family 27 subfamily a polypeptide 1
S100g	1483.02	−3.49	4.79 × 10^−2^	12,309	S100 calcium-binding protein G
Slc8a1	774.91	−3.61	1.76 × 10^−14^	20,541	solute carrier family 8 (sodium/calcium exchanger) member 1
Vdr	10,952.98	−2.13	1.73 × 10^−7^	22,337	vitamin D (1,25-dihydroxyvitamin D3) receptor

VAS-Inf comparison of infected VAS (*n* = 4) versus non-infected VAS (*n* = 3). The comparison included 4524 DEGs, of which 1424 were upregulated, 3100 downregulated, and 10,816 showed no change. The list order is ranked alphabetically by the official gene symbol. Only the DEGs related to VD are included here. For a full list of the DEGs reflecting VAS-Inf comparison in the colon dataset see list I in the Appendix A. **Abbreviations:** vitamin D receptor (VDR); vitamin A-sufficient (VAS); infection effect under the VAS status (VAS-Inf); differentially expressed gene (DEG); padj: DESeq2-computed adjusted *p*-value; base mean: mean expression level across all the 16 libraries, with DESeq2 normalization.

**Table 4 nutrients-14-01563-t004:** Ion transport-related DEGs altered by the *C. rodentium* infection in the colon.

Upregulated during Infection					
	Base Mean	Fold Change	padj	ENTREZID	Full Name
Adora2b	14.73	3.61	2.51 × 10^−3^	11,541	adenosine A2b receptor
Aqp4	24,562.26	5.41	6.42 × 10^−19^	11,829	aquaporin 4
Atp12a	21,642.30	2.38	1.36 × 10^−2^	192,113	ATPase H^+^/K^+^-transporting non-gastric alpha polypeptide
Atp1b2	24,979.73	21.05	1.90 × 10^−15^	11,932	ATPase Na^+^/K^+^-transporting beta 2 polypeptide
Atp2a2	14,010.80	2.36	1.84 × 10^−12^	11,938	ATPase Ca^++^-transporting cardiac muscle slow twitch 2
Car13	832.60	2.57	3.69 × 10^−13^	71,934	carbonic anhydrase 13
Car8	392.90	3.28	5.17 × 10^−12^	12,319	carbonic anhydrase 8
Clcn1	19.82	5.18	6.28 × 10^−6^	12,723	chloride channel voltage-sensitive 1
Clic1	11,306.30	2.58	6.81 × 10^−47^	114,584	chloride intracellular channel 1
Fosb	3418.09	8.33	2.72 × 10^−7^	14,282	FBJ osteosarcoma oncogene B
Slc9a3	695.50	2.75	8.18 × 10^−3^	105,243	solute carrier family 9 (sodium/hydrogen exchanger) member 3
**Downregulated during infection**					
Ano1	5571.44	−2.65	1.79 × 10^−10^	101,772	anoctamin 1 calcium-activated chloride channel
Ano2	23.72	−8.24	2.21 × 10^−8^	243,634	anoctamin 2
Aqp1	1252.79	−4.39	3.60 × 10^−11^	11,826	aquaporin 1
Aqp7	21.12	−10.28	5.10 × 10^−6^	11,832	aquaporin 7
Best2	1165.72	−3.66	2.27 × 10^−9^	212,989	bestrophin 2
Camk2a	143.61	−2.43	5.14 × 10^−6^	12,322	calcium/calmodulin-dependent protein kinase II alpha
Car11	65.93	−5.89	3.07 × 10^−16^	12,348	carbonic anhydrase 11
Car14	17.35	−13.72	1.41 × 10^−7^	23,831	carbonic anhydrase 14
Car15	99.56	−2.40	1.16 × 10^−5^	80,733	carbonic anhydrase 15
Car2	15,652.54	−2.06	1.24 × 10^−5^	12,349	carbonic anhydrase 2
Car3	2010.26	−4.40	4.29 × 10^−3^	12,350	carbonic anhydrase 3
Car4	18,070.39	−2.68	1.21 × 10^−2^	12,351	carbonic anhydrase 4
Cftr	3901.82	−2.43	2.21 × 10^−11^	12,638	cystic fibrosis transmembrane conductance regulator
Chp2	1689.23	−2.26	9.99 × 10^−7^	70,261	calcineurin-like EF hand protein 2
Slc15a1	2813.63	−5.26	7.65 × 10^−19^	56,643	solute carrier family 15 (oligopeptide transporter) member 1
Slc26a3	13,041.41	−2.04	2.25 × 10^−2^	13,487	solute carrier family 26 member 3
Slc9a2	3212.65	−2.22	1.25 × 10^−5^	226,999	solute carrier family 9 (sodium/hydrogen exchanger) member 2

VAS-Inf comparison of infected VAS (*n* = 4) versus non-infected VAS (*n* = 3). The comparison included 4524 DEGs, of which 1424 were upregulated, 3100 downregulated, and 10,816 showed no change. The list order is ranked alphabetically by the official gene symbol. Only the DEGs related to ion transport are included here. For a full list of the DEGs reflecting VAS-Inf comparison in the colon dataset see list I in the Appendix A. **Abbreviations:** differentially expressed gene (DEG); vitamin A-sufficient (VAS); infection effect under the VAS status (VAS-Inf); padj: DESeq2-computed adjusted *p*-value; base mean: mean expression level across all the 16 libraries, with DESeq2 normalization.

## Data Availability

The RNAseq datasets used in this study can be found in NCBI Gene Expression Omnibus under accession No. GSE143290.

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
