# Peer review of "Transcriptional Profiling of the Small Intestine and the Colon Reveals Modulation of Gut Infection with *Citrobacter rodentium* According to the Vitamin A Status"

_nutrients, 2022, doi:10.3390/nu14081563_

Round 1
Reviewer 1 Report
The manuscript by Chai et al., describes the results of an RNAseq study examining the effect of VA status and infection with C. rodentium on gene expression in the small intestine and colon. The authors describe changes in gene expression in the colon due to VA status and infection. They identify biological processes including epithelial hyperplasia, apoptosis, goblet cell depletion, T cell responses and others that are associated with infection in VA adequate mice. They also identify VA metabolic and VDR-related genes altered by infection as well as ion transport DEGs altered by infection. Colon DEGs were also identified that were modified by infection in VA deficient mice allowing for description of 4 categories of genes based on differences in the response to infection based on VA status. Finally, the authors presented co-expression network data generated by WGNCA for the traits VA or infection status. The data is compelling and supports and extends previous work in the fields of VA research and the effect of C. rodentium pathogenesis. The manuscript is well written but several points, however, need to be addressed.
1) It would be helpful to provide basic information on the composition of the diets so the reader does not need to refer to other publications. They are purified diets but are they based on an AIN-93 formulation?
2) Sections of the methods were taken verbatim from the previous manuscript describing some of the results from this study. Is this acceptable to the journal? May show up as plagiarism.
3) Supplemental Figure 1. The authors state the feces was plated on LB agar plates. The methods indicated LB/Nalidixic acid plates.
4) There is a surprising lack of expression of genes typically induced by a CR infection including IL17A, IL22 and IFNg. Others have shown IL-17A, IL-22, and IFN-g response around day 10 post-infection (e.g., Nat Med. 2011 Jun 12;17(7):837-44; Nat Med. 2008 Mar;14(3):282-9. How do the authors account for the lack of increased expression of these genes in response to C. rodentium infection?
5) Lines 394-95 state, “Scenario 2 comprised genes that were downregulated in the VAS-Inf comparison while being either higher or unchanged in the VAD-Inf comparison” While there is clear downregulation in the VAS-Inf comparison between uninfected and infected mice, I do not see how the VAD-Inf comparison can be interpreted as being higher in the infected mice compared to the uninfected mice.
6) The authors indicate that the 4 interaction Scenarios were subjected to GO and KEGG analysis but only GO enrichment is referred to in the text of Section 3.5. KEGG results are presented in Suppl. Figure 4 and Figure 5 for Scenarios 1 and 4, respectively. The text states that only a couple or no GO enrichment categories were identified for Scenarios 2 and 3, but what about KEGG pathways for those two Scenarios?
7) Line 662-remove the “and”
8) Line 667-should be “diarrhea”
9) Line 704- “Dehydration, rather than sepsis, is the major cause of lethality in the C. rodentium infection within the resistant mouse strains [29].” Should read susceptible not resistant.
10) The results discussed in section 3.6 are vague. The analysis shows correlations between the module eigengenes with the traits, VA status and Infection status but tells you nothing specific about the types of genes or pathways present in these modules. Nor does it tell you how many genes are associated with the modules, 10, 100 1000? The graphical results in Fig. 6b and c suggest different eigengenes trends between the Colon(Brown) and Colon(Purple) for VA status but how does this translate into something biological that sheds light on the genes in these modules and their relationship to the traits. The authors did a better job in describing the genes and their relative contribution to the modules in their previous publication of related results from the small intestine (Chai et al ., J Nutr. Biochem. 98 (2021) 108814). This section is the weakest part of the manuscript and needs to be significantly improved prior to publication.
11) It seems that other infection traits such as C. rodentium load, pathology index or crypt height would be amenable to WGNCA analysis as well. In this experiment were there any pathology changes or crypt height changes noted?
Author Response
Review 1
The manuscript by Chai et al., describes the results of an RNAseq study examining the effect of VA status and infection with C. rodentium on gene expression in the small intestine and colon. The authors describe changes in gene expression in the colon due to VA status and infection. They identify biological processes including epithelial hyperplasia, apoptosis, goblet cell depletion, T cell responses and others that are associated with infection in VA adequate mice. They also identify VA metabolic and VDR-related genes altered by infection as well as ion transport DEGs altered by infection. Colon DEGs were also identified that were modified by infection in VA deficient mice allowing for description of 4 categories of genes based on differences in the response to infection based on VA status. Finally, the authors presented co-expression network data generated by WGNCA for the traits VA or infection status. The data is compelling and supports and extends previous work in the fields of VA research and the effect of C. rodentium pathogenesis. The manuscript is well written but several points, however, need to be addressed.
Re
Response: We are very grateful for the kind words and constructive advice, according to which we have improved this manuscript remarkably. The reviewer has a very thorough understanding of this and relevant works in the past, including publication of ours and in this field. We are thankful to the reviewer because we were able to strengthened some key elements (e.g. WGCNA network interpretation, and key validation genes such as Il17a), clarified a few confusions/typos, and inspired for a few interesting future directions.
1) It would be helpful to provide basic information on the composition of the diets so the reader does not need to refer to other publications. They are purified diets but are they based on an AIN-93 formulation?
Response: Thank you, we are now providing a diet composition table in the supplementary Excel, so that the future readers won’t have to look for it from the original paper [1].
2) Sections of the methods were taken verbatim from the previous manuscript describing some of the results from this study. Is this acceptable to the journal? May show up as plagiarism.
Thank you so much for bringing this up, which was also pointed out by the journal editor. We took this issue very seriously, and tackled it via rephrasing, condensing, and citing our previous work. We focused on the Method session, and extended throughout the whole manuscript. Now we are confident to meet the journal’s requirement, that the “duplicate rate of the whole manuscript is less than 30%, and the single duplicate rate is less than 10%.”
3) Supplemental Figure 1. The authors state the feces was plated on LB agar plates. The methods indicated LB/Nalidixic acid plates.
Thank you so much for the suggestion. We indeed used LB agar plates containing 50 μg/mL nalidixic acid in our studies, to select and quantify the C. rodentium strain ICC169 (nalidixic acid resistant). Therefore, the method was correct. In the legend of supplementary Figure 1, we merely mentioned LB plate, but not nalidixic acid, with the intention of succinctness. But we agree with your suggestion and have added this detail in the legend of supplementary Figure 1, to make it unambiguous to the readers.
4) There is a surprising lack of expression of genes typically induced by a CR infection including IL17A, IL22 and IFNg. Others have shown IL-17A, IL-22, and IFN-g response around day 10 post-infection (e.g., Nat Med. 2011 Jun 12;17(7):837-44; Nat Med. 2008 Mar;14(3):282-9. How do the authors account for the lack of increased expression of these genes in response to C. rodentium infection?
We completely agree that those are important genes that should be elevated during C. rodentium infection. The reason they were not identified as DEG was their low abundance. “For the colon study, if the expression level was lower than 10 in more than 10 of 168 the samples, or if the sum of the expression level in all 16 samples was lower than 220, the transcript was regarded as a ‘low-expressed gene’ and removed.” With those criteria, out of the 24421 genes derived from mapping, 9081 low expression genes were removed, which contains Il17a, Il22, and Ifng. But when checking the raw data prior to filtering, the elevation trend of those three genes are obvious, especially when comparing VAS-uninf vs VAS-Inf groups (table below).
As a note, the filtering of low abundance gene upstream of differential expression analysis is necessary. Without this step the multiple correction burden will be even higher, making the differential expression analysis less powerful. Although those three cytokine’s mRNA were not included in the differential expression analysis, their receptors were included due to their relatively higher abundance, such as Il-17 receptors a through e, IL-22 receptors a1 and a2, as well as Ifngr1 and Ifngr2. This information, combined with other DEGs, led to the enrichment of “IL-17 signaling pathway,” “Th17 cell differentiation,” and “response to interferon-gamma,” which can indirectly serve as proof of principle that the cytokines were induced during peak infection.
We highly value this comment, and has added a paragraph in section 4.1, where we also cited the two seminal paper mentioned- “Some genes such as Il17a, Il22, and Ifng that would be expected to be increase were not detected as DEGs in 2.5our study [2,3], perhaps due to the filtering and expression-level criteria used. However, inspection of the raw data prior to filtering showed that there were changes in the expected direction in these genes. Moreover, the more abundant expressions of the receptors of those cytokines (Il-17 receptors a through e, IL-22 receptors a1 and a2, IFNg receptors Ifngr1 and Ifngr2), combined with other DEGs, helped to drive the enrichment categories of “IL-17 signaling pathway,” “Th17 cell differentiation,” and “response to interferon-gamma,” which can indirectly serve as proof of principle that the cytokines were induced during peak infection.”
5) Lines 394-95 state, “Scenario 2 comprised genes that were downregulated in the VAS-Inf comparison while being either higher or unchanged in the VAD-Inf comparison” While there is clear downregulation in the VAS-Inf comparison between uninfected and infected mice, I do not see how the VAD-Inf comparison can be interpreted as being higher in the infected mice compared to the uninfected mice.
Thank you so much for the question. Scenario 2 has 116 genes, which is comprised of 2 gene lists: 1) 110 genes downregulated in the VAS-Inf comparison while being unchanged in the VAD-Inf comparison, with some example genes shown in Fig 4. 2) 6 genes downregulated in the VAS-Inf comparison while being upregulated in the VAD-Inf comparison. Your question focuses on the latter, the 6 DEGs. As shown in the enclosed table below, we were indeed able to identify those 6 DEGs, that were downregulated in the VAS-Inf comparison (Log2FoldChange_VAS-Inf<-1, padj_VAS-Inf<0.05), while they are also upregulated DEGs in the VAD-Inf comparison (Log2FoldChange_VAD-Inf>1, padj_VAD-Inf<0.05).
We can understand the source of your confusion- those 6 genes are rare events, therefore only the top 6 rows in the heatmap corresponded to those genes (Scenario 2, Fig 3b), which can be hard to notice. The remaining rows of this heatmap reflects the 110 genes, that were downregulated in the VAS-Inf comparison while being unchanged in the VAD-Inf comparison. By “unchanged,” we mean those gene were identified as neither upregulated (Log2FoldChange >1, padj<0.05) nor downregulated DEG (Log2FoldChange <-1, padj<0.05)- note the stringent criteria of DEG used here led to a lot of “unchanged” genes.
Thanks to your thorough understanding of our work, and noticing the crucial details, we have added to the main text
- a) Definition of unchanged gene in section 2.5.
- b) Break down numbers of DEG count in Scenario 1 and 2, respectively, to minimize the possibility of future misunderstanding (see section 3.5).
6) The authors indicate that the 4 interaction Scenarios were subjected to GO and KEGG analysis but only GO enrichment is referred to in the text of Section 3.5. KEGG results are presented in Suppl. Figure 4 and Figure 5 for Scenarios 1 and 4, respectively. The text states that only a couple or no GO enrichment categories were identified for Scenarios 2 and 3, but what about KEGG pathways for those two Scenarios?
That is a great suggestion on language accuracy and rigorousness. Thank you! We checked section 3.5. When we say “no GO enrichment category” identified, it actually means “no GO or KEGG categories” identified. Since in most cases, when there’s no GO terms found, no KEGG terms would have been enriched. Therefore, when we say “no GO enrichment category” identified, it implied “no GO or KEGG categories” identified. We are grateful for you to point this out, because we agree that we should be more explicit, rather than relying on our readers’ subconscious understanding of this implication.
As a result, we have carefully reviewed section 3.5, and substitute “GO categories” with “functional categories.” This should be proper and succinct, since we had already defined in section 2.7 that our Functional enrichment methods contain two branches: GO and KEGG. We also checked the other part of the text, and make sure everything is correctly stated throughout.
7) Line 662-remove the “and”
Thank you so much for the suggestion. We have corrected this.
8) Line 667-should be “diarrhea”
Thank you so much. We followed the suggestion and improved the clarity.
9) Line 704- “Dehydration, rather than sepsis, is the major cause of lethality in the C. rodentium infection within the resistant mouse strains [29].” Should read susceptible not resistant.
Thank you so much for the suggestion, according to which we have re-visited the literature and corrected the misunderstanding we had earlier.
10) The results discussed in section 3.6 are vague. The analysis shows correlations between the module eigengenes with the traits, VA status and Infection status but tells you nothing specific about the types of genes or pathways present in these modules. Nor does it tell you how many genes are associated with the modules, 10, 100 1000?
The graphical results in Fig. 6b and c suggest different eigengenes trends between the Colon(Brown) and Colon(Purple) for VA status but how does this translate into something biological that sheds light on the genes in these modules and their relationship to the traits. The authors did a better job in describing the genes and their relative contribution to the modules in their previous publication of related results from the small intestine (Chai et al ., J Nutr. Biochem. 98 (2021) 108814). This section is the weakest part of the manuscript and needs to be significantly improved prior to publication.
Thank you for the great suggestion, according to which we have added
- a) 13 tabs in the supplementary Excel file, each containing the gene list corresponding to the modules
- b) Supplementary Figure 6, containing the remaining 10 eigengenes values that were not shown in Fig 6; bullet points a) and b) will allow readers to further explore the data.
- c) Supplementary Table 2, summarizing the gene numbers of each module, and the functional pathways each module was enriched in. We have also included more content in the Result section 3.6, to introduce the modules correlated with Infection status, and their enrichment pathways, content listed below
“…among which Colon (Blue, Yellow, Purple, and Greenyellow) modules were negatively correlated with the trait Infection status (Figure 6a). Among those four modules, Colon(Blue) module was most significant (p values=9×10-12, correlation coefficient= -0.98). The correlation coefficient was negative, meaning the module contained gene with overall lower expression levels in the infected colons (Figure 6a). Not surprisingly, among the 5674 module members, more than half of the genes (n=2967) were identified as the downregulated DEGs. Colon(Blue) module mainly exhibited functional enrichment in neurological functions, transport, and extracellular matrix (Supplementary Table 2), meaning those activities were reduced in the colon of infected mice, comparing with their non-infected counterparts. Colon(Turquoise) module was positively correlated with the Infection status and enriched for functional categories involved in mRNA processing, chromosome segregation, and protein catabolic process, etc (Supplementary Table 2)…. For each modules, module member gene list (supplementary Excel), and eigengene bar graph (Supplementary Figure 6) are provided.”
To further put the WGCNA more into biological contexts, the interpretation has been integrated in their designated Discussion sections, regarding those modules-
Section 4.1 “This is in line with the fact that Colon(Turquoise), a WGCNA module positively correlated with the Infection status, was enriched for biological functions such as chromosome segregation, and protein catabolic process (Supplementary Table 2).”
Section 4.2 “In addition, Colon (Blue) and Colon (Purple) were two WGCNA modules suggesting lower transmembrane transport activities in infected colon (Supplementary Table 2).”
11) It seems that other infection traits such as C. rodentium load, pathology index or crypt height would be amenable to WGNCA analysis as well. In this experiment were there any pathology changes or crypt height changes noted?
Thank you for the insight on suggesting additional parameters for testing against WGCNA module eigengenes. We completely agree that other parameters can be very informative, and as a matter of fact, we had performed similar analyses on body weight change percent, and gender, but ended up finding no significantly associated modules in colon.
Fecal shedding reflects the C. rodentium load in the mouse colon. We tested association between eigengenes and fecal shedding level. However, it turned out conveying almost the same information with that of the Infection_status. This is because the test result was mainly driven by the bifurcate difference between uninfected (shedding level=0) and infected (shedding level>0) status, as compared to the quantitative divergences between different infected individuals. In order to better mine genes responsible for the quantitative shedding levels, it would be better to construct an Infected-only WGCNA network, containing ideally more than 20 infected mice. With the small n in our current dataset (6 Infected VAD mice, and 4 Infected VAS mice), it would not be the most biologically meaningful analysis to do, but could serve as a future direction.
Using the same animal model, our group have already reported the Infection effect and VA effect regarding pathology index and crypt height in a previous paper (Fig 2, PMID: 25964475), demonstrating elevation of histology score and crypt length during peak of infection (p.i. day 10), reflecting inflammation and epithelial hyperplasia, respectively. The Infection-induced histological damage were further exacerbated under VAD status, compared to VAS (Fig 2, PMID: 25964475). Therefore, in the current RNAseq work, pathology index and crypt height were not part of our hypothesis. As a result, the colon tissue was snap frozen for RNA extraction, and not for histology. Nevertheless, we find your comment inspiring to correlate histopathology with mRNA expression profile. It could be a potential future direction to further dissect out a subset of genes responsible for the pathological severity.
Reviewer 2 Report
This interesting manuscript from Ross and colleagues provides an immense amount of new data obtained from RNAseq studies carried out in mice maintained on either a vitamin A-sufficient or a vitamin A-deficient diet in the face of GI infection with C. rodentium. The authors focused on the effects of these manipulations on gene expression in the lower small intestine and in the colon. Very thorough bioinformatics analyses of these expression data have allowed the authors to conclude that infection downregulates genes associated with the metabolism of both vitamins A and D, as well as dysregulation of genes involved in ion transport, including upregulation of chloride secretion and dysregulated bicarbonate metabolism. The authors argue that the disruptions of normal micronutrient metabolism and ion transport, together with a compromised immune response in vitamin A-deficient hosts may account for the increased morbidity and mortality under conditions of inadequate vitamin A intake.
The manuscript is well written. The topic of the study is an interesting one as it provides, at the transcriptional level, a comprehensive and better understanding of the increased incidence and severity of GI infections in vitamin A-deficient hosts, including children. The authors data are generally believable and the conclusions the authors reach from the data are generally appropriate. The authors’ new findings extend understanding of how vitamin A may act in preventing GI infections and in lessening their severity.
Although this is a strong contribution, there is one issue that the authors need to make clear in the text. The authors have chosen to study the lower portion of the small intestine and the colon and not the proximal small intestine where the majority of dietary vitamin A is absorbed and metabolized. One of the conclusions from this study is that vitamin A absorption and metabolism is dysregulated in C. rodentium infection. Data obtained from study of the colon clearly show this dysregulation. But how important is vitamin A absorption and metabolism in the colon compared to the proximal small intestine for maintaining normal whole body vitamin A physiology? Does it account for 5% or more of what can be attributed to the proximal small intestine? The authors need to put their finding regarding dysregulated expression of vitamin A metabolic genes into perspective. The text suggests that this is not observed for the distal small intestine tissue that was studied. Did the authors evaluate possible effects in the proximal small intestine? The present presentation of the new findings may be improperly interpreted by readers who are not familiar with vitamin A uptake and metabolism. The authors need to make clear to readers that the colon is not usually considered to be a major site for dietary vitamin A uptake or metabolism. One can appreciate the importance of changes to electrolyte balance and immune cell recruitment to the severity of the infection. But one must wonder whether effects on vitamin A and D metabolic genes are equally significant. The authors need to make a stronger case for this possibility if they wish to maintain the present treatment of this issue throughout the manuscript.
Author Response
Review 2
Although this is a strong contribution, there is one issue that the authors need to make clear in the text. The authors have chosen to study the lower portion of the small intestine and the colon and not the proximal small intestine where the majority of dietary vitamin A is absorbed and metabolized. One of the conclusions from this study is that vitamin A absorption and metabolism is dysregulated in C. rodentium infection. Data obtained from study of the colon clearly show this dysregulation.
But how important is vitamin A absorption and metabolism in the colon compared to the proximal small intestine for maintaining normal whole body vitamin A physiology? Does it account for 5% or more of what can be attributed to the proximal small intestine? The authors need to put their finding regarding dysregulated expression of vitamin A metabolic genes into perspective.
We appreciate the reviewer’s kind recognition of our work, and agree that it is very important to reiterate small intestine is the major organ for VA absorption, especially for future readers who are not familiar with VA uptake and metabolism.
Initially, we also found the colon expression somewhat surprising. Although it is well established that the major portion of vitamin A metabolism occurs higher up in the intestine, we do not know that colon absorption can be ruled out, especially considering the recent findings that water soluble vitamin can be efficiently absorbed in colon [4]. Rodent colon does contain VA (Fig 3D of [5]), indicating there is either direct absorption in colon and/or some vitamin A is absorbed in small intestine then allocated to colon via blood circulation. Our observation on colon gene profiles suggested that VA may be metabolized to retinoic acid locally, the rate of which could influence the health of the colonocytes, or the normal functions of immune cells in the lamina propria. At present we cannot provide further insight into these observations, but have added notes in section 4.3 to remind the future readers that SI remains the major organ for VA absorption, and that VA re-distributed to colon may play crucial role locally.
The text suggests that this is not observed for the distal small intestine tissue that was studied. Did the authors evaluate possible effects in the proximal small intestine? The present presentation of the new findings may be improperly interpreted by readers who are not familiar with vitamin A uptake and metabolism.
The authors need to make clear to readers that the colon is not usually considered to be a major site for dietary vitamin A uptake or metabolism. One can appreciate the importance of changes to electrolyte balance and immune cell recruitment to the severity of the infection. But one must wonder whether effects on vitamin A and D metabolic genes are equally significant. The authors need to make a stronger case for this possibility if they wish to maintain the present treatment of this issue throughout the manuscript.
To clarify, as we published in an earlier paper (PMID: 34242724), VA effect was observed in lower small intestine (18 upregulated, and 31 downregulated DEGs when comparing VAS SI vs VAD SI), which is in line with the fact that small intestine is the major site of VA absorption. However, the current paper intends to focus on the Infection effect and the Interaction effect. The differential expression analysis identified 0 DEGs corresponding to the VAS-Inf or the Interaction effect in lower small intestine, therefore we stated those effects were only observed in colon, not small intestine (End of section 3.2), which does not conflict with small intestine being the major organ of VA absorption.
In order to capture all three effects (VA, Infection, and Interaction), we chose lower small intestine as a representation of small intestine. This decision is based on the C. rodentium kinetics (PMID: 25964475), and the premise that the small intestine is the primary site of VA absorption. However, it’s not very surprising that Infection and Interaction effects were not identified by differential expression in lower small intestine- the passing through of C. rodentium in lower small intestine may be too transient, and masked by the lack of synchronization and/or individual variation between different animals, thus it may not cause large enough differences to be detected statistically.
Proximal small intestine was not examined by bulk RNAseq in this series of study, but this could be a future direction. Based on our current data, we speculate the proximal, or upper small intestine may behave very similarly to the lower small intestine: more DEGs related to VA effect as compared to that in lower small intestine, and 0 DEG related to Infection or Interaction. Therefore, even if we had performed RNAseq using the upper small intestine, we expect it will most likely fall into the scope of the VA effect, as addressed in our previous publication (PMID: 34242724), rather than this current one.
References
- Smith, S.M.; Hayes, C.E. Contrasting impairments in IgM and IgG responses of vitamin A-deficient mice. Proceedings of the National Academy of Sciences 1987, 84, 5878-5882.
- Geddes, K.; Rubino, S.J.; Magalhaes, J.G.; Streutker, C.; Le Bourhis, L.; Cho, J.H.; Robertson, S.J.; Kim, C.J.; Kaul, R.; Philpott, D.J. Identification of an innate T helper type 17 response to intestinal bacterial pathogens. Nature medicine 2011, 17, 837-844.
- Zheng, Y.; Valdez, P.A.; Danilenko, D.M.; Hu, Y.; Sa, S.M.; Gong, Q.; Abbas, A.R.; Modrusan, Z.; Ghilardi, N.; De Sauvage, F.J. Interleukin-22 mediates early host defense against attaching and effacing bacterial pathogens. Nature medicine 2008, 14, 282-289.
- Said, H.M. Recent advances in transport of water-soluble vitamins in organs of the digestive system: a focus on the colon and the pancreas. American Journal of Physiology-Gastrointestinal and Liver Physiology 2013, 305, G601-G610.
- Li, Y.; Wei, C.-H.; Hodges, J.K.; Green, M.H.; Ross, A.C. Priming with Retinoic Acid, an Active Metabolite of Vitamin A, Increases Vitamin A Uptake in the Small Intestine of Neonatal Rats. Nutrients 2021, 13, 4275.
Round 2
Reviewer 1 Report
The authors adequately responded to my comments and the manuscript is much improved.